# COMPLEX-VALUED SCATTERING REPRESENTATIONS

## ABSTRACT

Complex-valued deep learning has made significant progress with manifold geometry and group theory. It delivers leaner and better classifiers with novel complex-valued layer functions and network architectures, not only on naturally complex-valued data such as Magnetic Resonance imaging (MRI) but also on real-valued data such as RGB or multi-spectral images. However, current complex-valued representations for complex-valued and real-valued inputs are rudimentary, focusing on channel characteristics (*e.g., sliding encoding*) without capturing spatial and spatial-frequency properties of the input data. We propose Complex-valued Scattering Representations (CSR) as universal complex-valued representations and integrate them into complex-valued deep learning networks. To obtain CSR, We construct filters based on complex-valued Morlet wavelets with tunable parameters and develop learnable high-dimensional complex-valued ReLU as the nonlinear activation function. By incorporating these novel components into complex-valued models, our models significantly outperform real-valued counterparts and existing complex-valued models on RGB, multi-spectral image (MSI), and MRI patch classification tasks, especially under *limited labeled* training data settings, greatly enhancing complex-valued networks on a broader range of applications.

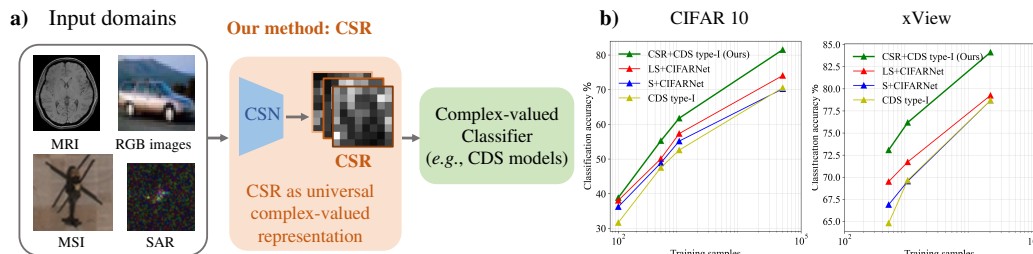

Figure 1: **Complex-valued Scattering Representations (CSR) serve as universal complex-valued representations for a wide range of input domains. a):** Given image from an input domain (*e.g., RGB image, MRI, MSI*), our Complex-valued Scattering Networks (CSN) extract the complex-valued scattering representations, which are then fed into the complex-valued classifiers. **b):** By incorporating CSR with Complex-valued deep learning models (Here we use CDS from Singhal et al. (2022a)), our approaches significantly outperform CDS and other real-valued counterparts with different training samples on CIFAR 10 and xView benchmarks.

## 1 INTRODUCTION

Complex-valued deep learning has emerged as a powerful approach to modeling complex-valued data, leveraging the unique algebraic operations and properties of complex-valued data to develop more accurate and efficient models. Recent developments in manifold geometry (Chakraborty et al., 2020; Singhal et al., 2022a) and group theory have further advanced the field, leading to the creation of leaner and better classifiers with novel complex-valued layer functions and network architectures (Virtue et al., 2017; Trabelsi et al., 2017; Singhal et al., 2022a; Chakraborty et al., 2020).

While complex-valued deep learning was initially developed to better model naturally complex-valued data such as Magnetic Resonance Imaging (MRI), and Synthetic Aperture Radar (SAR), it has recently been shown to be effective for real-valued input data such as RGB (Singhal et al., 2022a) or multispectral images (Singhal et al., 2022b) through complex-valued representations, delivering

leaner and better classifiers with novel complex-valued layer functions and network architectures. Singhal et al. (2022a) introduced "sliding" encoding to convert RGB color space to complex-valued representations. "Sliding" encoding maps adjacent color channels to the real and imaginary parts of a complex-valued channel to exploit the inter-channel correlations. These developments suggest the exciting possibility of a single model that can handle *both real and complex-valued data* and exploit the corresponding properties appropriately.

Despite notable progress achieved with current encoding methods, the resulting complex-valued representations for complex and real-valued inputs are rudimentary. Notably, these representations only model *channel* characteristics, lacking the ability to model any spatial and spatial-frequency properties of the input data. This is especially limiting because certain spatial-frequency properties, such as aliasing and off-resonance effects in MRI, play important roles in image recognition.

A promising solution that can capture both spatial and spatial-frequency features at the same time is the Wavelet transform. Wavelet transforms achieve this by using a set of filters that can decompose an image into different spatial-frequency bands at different scales, allowing for simultaneous extraction of both spatial and spatial-frequency features. Building on this property, Bruna & Mallat (2013) proposed real-valued Wavelet Scattering Networks (WSNs), which have achieved notable success in extracting non-learned features for image classification tasks, particularly when training with _limited labeled_ data (Oyallon et al., 2018; Gauthier et al., 2022).

Inspired by Wavelets and WSNs, here we propose learnable Complex-valued Scattering Representations (CSR) as a universal complex-valued representation to model the spatial and spatial-frequency properties of the input data. We introduce the term Complex-valued Scattering Networks (CSNs) to refer to the networks that produce CSR as their output for convenience. As shown in Figure 1, we further integrate CSR with complex-valued deep learning models, such as complex-valued Co-domain Symmetry models (CDS) (Singhal et al., 2022a), for downstream image classifications. As visualized in Figure 2, we construct filters based on complex-valued Morlet wavelets

We integrated CSR into complex-valued models (Linear Layer (LL) and CDS) and achieved significant classification performance improvements compared to CDS and other real-valued WSN-based models, especially on tasks with _limited labeled_ data. Our evaluation includes various benchmarks from different domains such as CIFAR 10/100 (Krizhevsky et al., 2009), xView MSI classification (Singhal et al., 2022b), and a newly introduced complex-valued MRI Patch classification dataset.

To summarize, we make the following contributions:

- We propose CSR, a universal complex-valued representation for extracting spatial and spatial-frequency features from diverse input domains in complex-valued deep learning.
- We introduce a novel learnable high-dimensional Complex-valued ReLU function as the non-linear activation module for our CSR. This module enhances the network's ability to adapt to the complexities of the input data effectively.
- By integrating CSR with complex-valued models, our approach outperforms complex-valued models and real-valued WSNs in CIFAR10/100, xView MSI, together with a new evaluation benchmark of complex-valued MRI patch classification.

We will publicly release our code and our new MRI patch classification dataset upon publication.

## 2 RELATED WORK

### 2.1 COMPLEX-VALUED NETWORKS

Complex-valued neural networks (CVNNs) are an extension of traditional real-valued neural networks designed to handle complex-valued data. Due to the importance of complex numbers in engineering and scientific disciplines (Needham, 1998), CVNNs have been an active topic since the early days of deep learning research. Nitta (2003b) analyze CVNNs in the context of the XOR problem and finds that the real and imaginary components of the decision boundary of a CVNN are orthogonal. Further works demonstrate better optimization properties (Nitta, 2002) and representational capacity (Nitta, 2003a). We refer the reader to Bassey et al. (2021) for a deeper review of CVNNs. A central question in this literature is how to adapt real-valued deep learning to complex numbers.

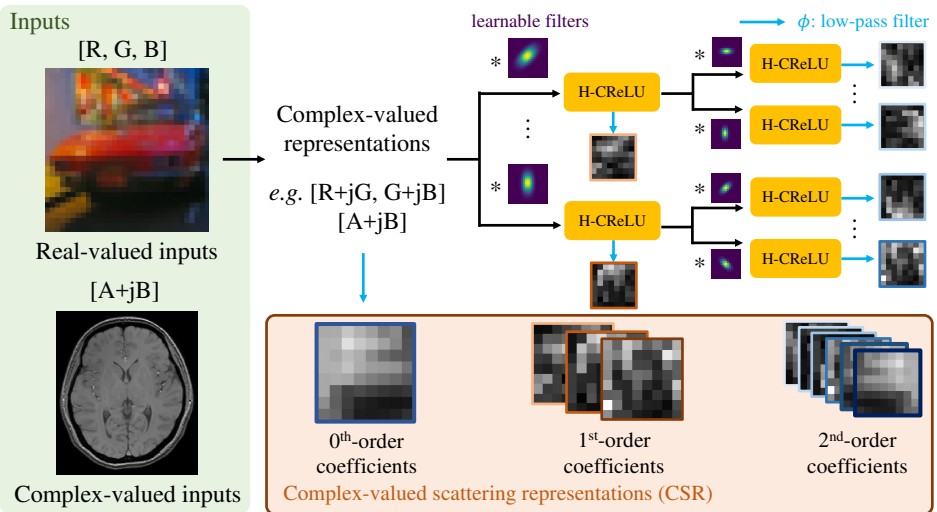

Figure 2: **Diagram of constructing CSR from input images.** Real-valued inputs (*e.g.,* RGB) are first converted to complex-valued representations using "Sliding" encodings (Singhal et al., 2022a). Complex-valued inputs, on the other hand, remain unchanged. We then convolve with learnable filters and apply our H-CReLU module to extract the scattering coefficients up to $2^{nd}$ order. H-CReLU lifts a complex number to high-dimensional space, applies point-wise CReLU, and maps back to a complex number. Coefficients from different orders are then concatenated to form CSR.

Previous works (Virtue et al., 2017; Zhang et al., 2017; Trabelsi et al., 2017) redefine basic building blocks for complex-valued networks, such as complex-valued convolution, batch normalization, and non-linear activations. However, those methods are not robust against complex-valued scaling. SurReal (Chakraborty et al., 2020) addressed this issue by modeling the complex value space as a manifold to enable robustness to complex-valued scaling. Meanwhile, Singhal et al. (2022a) developed equivariant and invariant neural network layers for co-domain transformation that outperform other complex-valued networks in image classification tasks.

While current complex-valued encoding approaches have made notable progress, they still lack the ability to effectively model both spatial and spatial-frequency properties of the input data. Our proposed CSR, on the other hand, successfully captures both types of features.

## 2.2 SCATTERING REPRESENTATIONS

Scattering representations, as proposed by Bruna & Mallat (2013), leverage pre-determined wavelet filters to create powerful hierarchical representations and extract features from both the spatial and spatial-frequency domains. From a mathematical perspective, these representations satisfy translation invariance up to a particular scale and are stable to deformations, making them a simple yet effective tool for signal analysis in various fields such as image and audio processing (Bruna & Mallat, 2013; Andén & Mallat, 2014; Hirn et al., 2015; Eickenberg et al., 2018). Benefiting from the well-designed filters, scattering representation-based models have shown promising results in applications with *limited labeled* data.

Oyallon et al. (2018) introduce hybrid networks, demonstrating the effectiveness of scattering transforms as early layers of learned CNNs. McEwen et al. (2021) constructed scattering networks on the sphere, providing a powerful representational space for spherical data. Gauthier et al. (2022) learns the geometric parameters of wavelet filters (*e.g.* orientation, aspect ratio), achieving new state-of-the-art results in a low-data regime.

Our CSR can be seen as an extension of real-valued scattering representations to the complex-valued domain, providing a universal and powerful representation for complex-valued deep learning.

## 3 METHOD

### 3.1 COMPLEX-VALUED SCATTERING REPRESENTATIONS

Figure 2 visualizes how we construct CSR from both real-valued and complex-valued inputs. For simplicity, we limit our focus to 2D CSNs and only consider their up to $2^{nd}$ order coefficients (Bruna & Mallat, 2013). For a real-valued image $I$ of $m$ channels, we first turn it into a complex-valued image of $m - 1$ channels through "sliding" color encoding (Singhal et al., 2022a):

$$I(u) = [I_1, I_2, ..., I_m] \rightarrow [I_1 + jI_2, I_2 + jI_3, ..., I_{m-1} + jI_m], \tag{1}$$

where $u$ is the spatial position index, $j = \sqrt{-1}$.

Our CSN starts with the complex-valued representation $I(u)$, a scaling integer $J \in \mathbb{N}$, and an integer $L \in \mathbb{N}$ representing the number of wavelet angular orientations. CSN computes the scattering coefficients $S^0 I$, $S^1 I$, and $S^2 I$ of orders 0, 1, and 2, respectively, which can be interpreted as the result of convolving $I(u)$ with 0, 1, and 2 wavelet filters. $J$ represents the spatial scale of the scattering transform.

As shown in Figure 2, to compute the $0^{th}$ order coefficient, we use a low pass filter $\phi_J$ with a spatial window of scale $2^J$ (here is the Gaussian smoothing function). To obtain the coefficient, we convolve the input signal $I(u)$ with $\phi_J$, and then downsample the result by a factor of $2^J$. This operation can be expressed as $S^0 I(u) = I * \phi_J(2^J u)$. To recover the high-frequency information that $S^0$ discards, higher-order coefficients are introduced using wavelets.

A Morlet wavelet family is derived by scaling and rotating a complex-valued mother wavelet $\psi$. Specifically, we obtain a particular Morlet wavelet at scale $j \geq 0$, rotation $\theta$, and aspect ratio $\gamma$ by dilating the mother wavelet as follows:

$$\psi_{j,\theta,\gamma}(u) = \frac{1}{2^{2j}} \psi_\gamma(r^{-\theta} \frac{u}{2^j}), \tag{2}$$

where $r^{-\theta}$ represents the rotation by $-\theta$. For real-valued SNs, it's important to note that the spatial-frequency domain exhibits conjugate symmetry. As a result, the rotation angle $\theta$ is constrained to range from $[0, \pi)$. In CSNs, we design $\theta$ to range from $[0, 2\pi)$.

To compute the $1^{st}$-order scattering coefficients, we convolve the input signal with one of the complex-valued wavelets $\psi_{j_i,\theta_i,\gamma_i}(u)$ and downsample the response by the scale $2^{J-j_i}$. Next, we apply a pointwise activation function $f(\cdot)$ to the downsampled signal to add nonlinearity. Finally, the smoothed signal is obtained by convolving it with the low-pass filter $\phi_J(2^J u)$. For real-valued SNs, $f(\cdot)$ is usually a complex modulus, which takes the absolute value $|\cdot|$ of a complex number. However, complex modulus discards its phase information which can be crucial for complex-valued applications where phase carries important information. Here, we propose a learnable activation function $f_w(\cdot)$, where $w$ is the learnable parameters (§ 3.2). Mathematically, the $1^{st}$-order coefficients can be expressed as:

$$S^1 I(u) = f_w(I * \psi_{j_i,\theta_i,\gamma_i}) * \phi_J(2^J u). \tag{3}$$

Similarly, as illustrated in Figure 2, we perform a second wavelet transform on each channel of the $1^{st}$-order coefficients before applying the low-pass filter. This can be written as:

$$S^2 I(u) = f_w(f_w(I * \psi_{j_i,\theta_i,\gamma_i}) * \psi_{j_k,\theta_k,\gamma_k}) * \phi_J(2^J u), \tag{4}$$

where $\psi_{j_k,\theta_k,\gamma_k}$ is the second filter we apply. Due to the spatial-frequency supports of filters, only coefficients with $j_i < j_k$ have significant energy (Bruna & Mallat, 2013).

Motivated by Gauthier et al. (2022), we let the network learn each wavelet's orientation $\theta$ and aspect ratio $\gamma$ to enable better adaptions to particular datasets. $\theta$ is initialized to be equally spaced on $[0, 2\pi]$, while $\gamma$ is initialized as a constant $\frac{4}{L}$. We adapted the Kymatio software package (Andreux et al., 2020) to implement CSNs.

Given the fact that the wavelet transforms are stable to deformations and $f_w(\cdot)$ being a point-wise function, following the proof in Bruna & Mallat (2013), we can derive that our CSR is stable to deformation and invariant to local translations.

## 3.2 LEARNABLE HIGH-DIMENSIONAL COMPLEX ReLU

As we pointed out, the complex modulus for SNs discards the important phase information from the signal. One alternative is to use complex-valued ReLU (CReLU) (Agarap, 2018; Trabelsi et al., 2017). However, CReLU destroys the phase information other than the first quadrant. Thus, instead of using a hand-crafted function, we proposed a learnable high-dimensional CReLU (H-CReLU) module (Orange block in Figure 2).

Motivated by other high-dimensional lifting methods (Suykens, 2001; Sandler et al., 2018), H-CReLU operates on a complex number $z \in \mathbb{C}$ by first lifting it to a higher-dimensional space using linear mapping. Specifically, we use a trainable matrix $\text{UP}_{N_h} \in \mathbb{C}^{N_h \times 1}$ to transform $z$ into a $N_h$-dimensional representation, where $N_h$ is set to 16 in our experiments. After lifting the input, we apply point-wise CReLU to the high-dimensional intermediate results. Finally, we map the high-dimensional intermediate results back to the original space using a trainable matrix $\text{DOWN}_{N_h} \in \mathbb{C}^{1 \times N_h}$. The resulting activation function, $f_w(z)$, can be then written as:

$$f_w(z) = \text{DOWN}_{N_h} \cdot \text{CReLU}(\text{UP}_{N_h} \cdot z), \tag{5}$$

where $\{\text{UP}_{N_h}, \text{DOWN}_{N_h}\}$ are the learnable matrices with $2N_h$ complex-valued learnable parameters. Ablation studies demonstrate the superior effectiveness of H-CReLU as $f_w(z)$.

## 3.3 CSR FOR DOWNSTREAM IMAGE CLASSIFICATION

We integrate CSR with complex-valued models for downstream image classification tasks. In our experiments, we integrate CSR with two types of well-established complex-valued networks: 1) Complex-valued linear layer; 2) Type-I CDS with CIFARNet architecture from Singhal et al. (2022a). We also include CDS-Large with Wide Residual Network (WRN) (Zagoruyko & Komodakis, 2016) architecture from Singhal et al. (2022a) for comparisons.

# 4 EXPERIMENTS

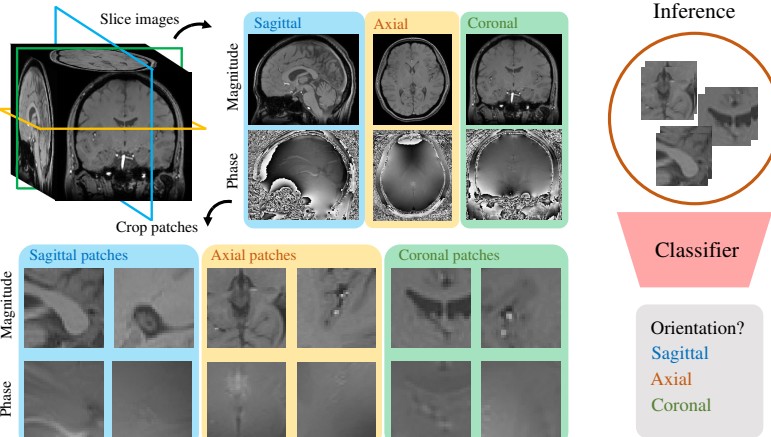

Figure 3: **Construction of complex-valued MRI patch classification dataset.** We start from complex-valued 3D MRI volumes obtained from Shi et al. (2022). Then, we sliced 2D images and cropped patches from different anatomical orientations (*i.e.,* sagittal, axial, coronal). The objective is to train a classifier that can correctly identify the anatomical orientation of the input patch.

We compare the performance of our approach against real-valued scattering representations and previous complex-valued models (without scattering) on four diverse image datasets: CIFAR 10, CIFAR 100, xVIEW MSI (Singhal et al., 2022b; Lam et al., 2018), and a newly introduced dataset for MRI patch classification. We also evaluate the performances under _limited-labeled_ training data. CIFAR 10 and CIFAR 100 are well-established natural RGB image classification benchmarks (Gauthier et al., 2022; Bruna & Mallat, 2013; Oyallon et al., 2018). xView MSI is a large-scale 8-band MSI dataset. Each channel within an 8-band image contains measurements obtained from a different

electromagnetic spectrum. Following Singhal et al. (2022b), xView consists of 60 total classes, from which we select 10 supercategories.

**MRI classification dataset** We created a new complex-valued dataset for MRI patch classification to showcase the effectiveness of CSR on naturally complex-valued data. To tackle the scarcity of labeled MRI data, we performed automatic labeling by slicing a volumetric MRI dataset into its cross-sections. We used complex-valued multi-echo 3D MRI volume data from Shi et al. (2022) to create our MRI patch dataset. The dataset includes 144 3D scans from eight healthy subjects. We take the first echo volumes and slice 2D images from three different orientations (*i.e.*, sagittal, axial, coronal). Then, as shown in Figure 3, 2D patches ($32 \times 32$) are extracted for each orientation. The objective is to train a classifier that can correctly identify the orientation of the complex-valued patch (3-classes classification task). Our training set consists of 26,640 patches extracted from 4 subjects, while the testing set consists of 17,520 patches from another 4 subjects.

We evaluate CSR by utilizing them with two common complex-valued models. In the first model, we considered CSR as the input of a simple LL. This configuration of LL helps us understand the linear separability of CSR. In the second case, we integrate CSR with recently proposed complex-valued CDS networks (Singhal et al., 2022a), where we experimented on Type-I CDS. For both models (LL and CDS), we compare our CSNs with their real-valued counterparts, including conventional scattering (S) and recently proposed learnable scattering (LS) (Gauthier et al., 2022). We design the networks to have a similar number of parameters for fair comparisons. For reference, we also compare our approach to CDS-Large (complex-valued) and WRN-16 (real-valued).

All of our models are implemented in PyTorch (Paszke et al., 2019) and optimized using AdamW (Kingma & Ba, 2014; Loshchilov & Hutter, 2017) with an initial learning rate of $3 \times 10^{-3}$, decayed by a factor 0.3 every 10k iterations. We use a batch size of 256 for 50k training iterations.

## 4.1 BENCHMARK COMPARISONS

| Method | CIFAR 10 | | | | CIFAR 100 | | | |
|---|---|---|---|---|---|---|---|---|
| - | 100 samples | 500 | 1000 | All | 1000 samples | 5000 | 10000 | All |
| *Scattering + Linear layers* | | | | | | | | |
| S (Bruna & Mallat, 2013)+LL | 35.78±0.62 | 48.32±0.30 | 53.52±0.24 | 65.46 | 17.03±0.74 | 33.00±0.50 | 37.98±0.22 | 41.12 |
| LS (Gauthier et al., 2022)+LL | 37.87±0.55 | 52.88±0.26 | 56.94±0.20 | 69.68 | 18.96±0.71 | 33.95±0.63 | 39.83±0.18 | 43.65 |
| CSR+LL † | **39.84** ±0.54 | **56.23** ±0.32 | **60.01**±0.16 | **74.30** | **20.07** ±0.82 | **34.54**±0.49 | **41.18**±0.29 | **47.81** |
| *Scattering + CIFARNet* | | | | | | | | |
| S (Bruna & Mallat, 2013)+CIFARNet | 36.23±0.70 | 48.88±0.54 | 55.17±0.18 | 70.23 | 17.29±0.93 | 30.44±0.39 | 36.23±0.28 | 40.76 |
| LS (Gauthier et al., 2022)+CIFARNet | 38.06±0.68 | 50.92±0.58 | 57.34±0.26 | 74.07 | 17.90±0.85 | 32.05±0.51 | 38.45±0.30 | 42.81 |
| CSR+CDS type-I † | **38.87**±0.49 | **55.26**±0.45 | **61.78** ±0.14 | **81.52** | **18.68**±0.77 | **34.24**±0.40 | **40.03**±0.37 | **46.80** |
| *CDS and large models (no scattering)* | | | | | | | | |
| CDS type-I (Singhal et al., 2022a) | 31.67±0.50 | 47.53±0.21 | 52.57±0.31 | 70.55 | 15.52±1.01 | 29.77±0.36 | 33.98±0.23 | 37.14 |
| CDS large (Singhal et al., 2022a) | **33.32**±0.98 | **48.65**±0.27 | **60.23**±0.13 | 93.27 | **17.30**±0.65 | 33.73±0.72 | 48.19±0.33 | 71.03 |
| WRN-16 (Zagoruyko & Komodakis, 2016) | 32.55±1.13 | 44.19±0.83 | 59.57± 0.40 | **96.34** | 17.03±1.38 | **36.99** ±1.04 | **53.98** ±0.57 | **76.35** |

† ours ; S: Scattering (Bruna & Mallat, 2013); LS: Learnable Scattering (Gauthier et al., 2022); CSR: Complex-valued Scattering Representations (ours).

\# Parameters for CIFAR 10 (CIFAR 100): 156k (1.6M) for S+LL; 156k (1.6M) for LS+LL; 207k (2.1M) for CSR+LL; 124k (136k) for S+CIFARNet; 124k (136k) for LS+CIFARNet; 122k (145k) for CSR+CDS type-I; 105k (128k) for CDS type-I; 1.7M (1.8M) for CDS large; 17.1M (22.4M) for WRN-16.

Table 1: **Classification accuracy for CIFAR 10 and CIFAR 100 benchmarks (mean $\pm$ std.).** We report results from models trained with varying sample sizes to demonstrate the effectiveness of CSRs. **Bold** highlights the best results in each category, while **Bold** represents the best results across all categories. CSRs outperform their real-valued counterparts and CDS in all training setups.

**CIFAR 10 (and CIFAR 100)** consists of 10 (100) classes containing 6,000 (600) images from each class. Each image has a size of $32 \times 32$. Both datasets are split into a training set of 50,000 images and a test set of 10,000 images. Additionally, we also evaluate the performance of CSRs in small data regimes with *limited labeled* data. To account for the randomness in data selection, we train the same model using ten different seeds for the small-size experiments. We evaluate training size of $\{100, 500, 1000, 50k\}$ for CIFAR 10, and $\{1000, 5000, 10000, 500k\}$ for CIFAR 100.

| Method | xView | | | MRI patch classification | |
|---|---|---|---|---|---|
| - | 500 samples | 1000 | All | 100 samples | 500 |
| *Scattering + Linear layers* | | | | | |
| S (Bruna & Mallat, 2013) + LL | 62.55±2.35 | 68.45±1.47 | 74.30 | 56.79±0.88 | 68.95±0.34 |
| LS (Gauthier et al., 2022) + LL | 67.69±2.01 | 71.14±1.88 | 75.78 | 67.03±0.64 | 85.40±0.42 |
| CSR + LL † | **71.83**±2.70 | **74.86**±1.1s7 | **80.04** | **74.22**±0.57 | **91.73**±0.33 |
| *Scattering + CIFARNet* | | | | | |
| S (Bruna & Mallat, 2013) + CIFARNet | 66.88±2.65 | 69.54±1.60 | 78.68 | 59.62±0.80 | 83.53±0.96 |
| LS (Gauthier et al., 2022) + CIFARNet | 69.49±2.05 | 71.72±1.68 | 79.25 | 71.86±0.98 | 94.74±0.60 |
| CSR + CDS type-I † | 73.07 ±1.79 | **76.18** ±1.21 | 84.13 | 84.80 ±1.06 | **99.18** ±0.15 |
| *CDS and large models* | | | | | |
| CDS type-I (Singhal et al., 2022a) | 64.80±2.45 | 69.65±1.33 | 78.69 | 54.49±0.34 | 69.77±0.38 |
| CDS large (Singhal et al., 2022a) | **68.45**±2.32 | **72.77**±0.98 | 81.80 | **82.68**±0.43 | 98.45±0.17 |
| WRN-16 (Zagoruyko & Komodakis, 2016) | 61.13±3.74 | 70.46±1.52 | **84.25** | 39.25±1.43 | 55.66±2.35 |

† ours ; S: Scattering; LS: Learnable Scattering; CSN: Complex-valued Scattering Network (ours).
# Parameters for xView (MRI Patch classification): 415k (31k) for S+LL; 415k (31k) for LS+LL; 726k (31k) for CSR+LL; 364k (76k) for S+CIFARNet; 364k (76k) for LS+CIFARNet; 357k (73k) for CSR+CDS type-I; 111k (102k) for CDS type-I; 1.8M (1.7M) for CDS large; 17.1M (17.1M) for WRN-16

Table 2: **Classification accuracy for xView and MRI patch classification dataset (mean ± std.).** XView models were trained with sample sizes of 500, 1000, and full size, while MRI patch classification models used 100 and 500 samples. For both datasets, our CSRs significantly outperform their real-valued counterparts. Table layouts and symbols are the same as Table 1.

Table 1 summarizes the results under different training setups. In the first LL category, our proposed CSR+LL outperforms the previous state-of-the-art LS method under all training setups. CSR+LL achieves a $> 4\%$ accuracy gain for the full-size ($50k$) training and sets a new state-of-the-art for small data training regimes. For the second CIFARNet comparison, Our proposed approach significantly outperforms its real-valued counterparts and the CDS model in all the comparisons. Besides, we also present the results of CDS type-I, CDS large (Singhal et al., 2022a), WRN-16 (Zagoruyko & Komodakis, 2016) as references and comparisons. While large networks tend to excel when trained on ample amounts of data, they often fall short when the available data is limited. In such scenarios, scattering-based methods tend to yield superior results.

**xView MSI dataset** Multi-band MSI remote sensing images consist of multiple bands in addition to RGB color images. xView MSI dataset contains a total of 86,980 images (size $32 \times 32$), with 20,431 images for training, 2,270 images for validation, and 63,279 images for testing. We use a spatial scale $J = 2$ and compare models trained with $\{500, 1000, \text{full size}\}$ samples. Our experimental results (shown in Table 1) demonstrate that CSRs consistently outperform real-valued networks and CDS without CSR across all training settings by a substantial margin. Furthermore, we found that our CSR+CDS model achieves the same level of accuracy as WRN-16 on full-sized training data while using only 2% of the parameters.

**MRI patch classification** Previous sections evaluated CSR on real-valued benchmarks. Here, we evaluate CSR on our complex-valued MRI patch classification dataset. MRI patch classification is typically considered easier than natural image classification tasks primarily due to the lower complexity and diversity of the data. Thus, we create two small-data training regimes: (1) using 100 samples from a single scan and (2) using 500 samples from 5 scans. Table 2 shows that our CSRs significantly outperform their real-valued counterparts, CDS (without CSR), and achieve better results compared with large models (*i.e.,* WSN and CDS large). It's noteworthy that, given the large network capacity and inadequate training data, WRN-16 performs poorly on this task.

## 4.2 Understanding CSR

To gain a better understanding of CSR, we analyze the learnable filters. Figure 4 showcases the visualization of data-specific scattering filters of CSNs in Fourier space that were trained with linear classification layers. The filters displayed in the figure were trained on CIFAR 10/100, xView, and

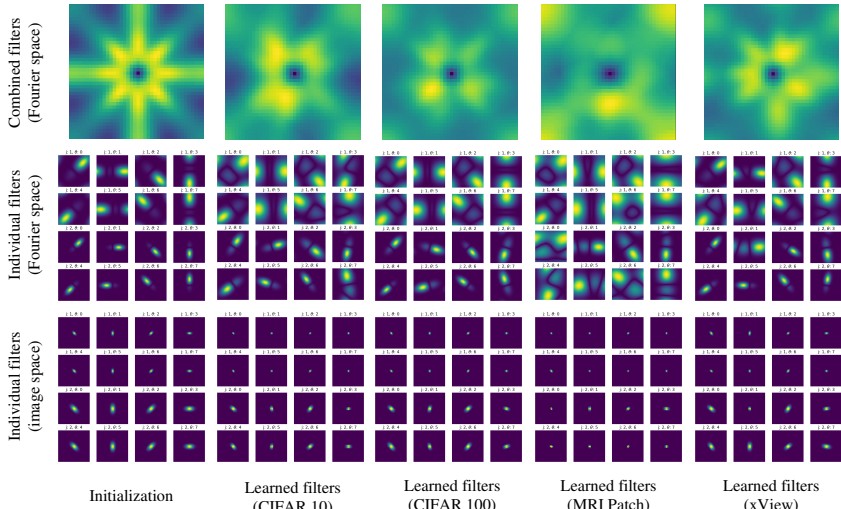

Figure 4: **Visualization of learned data-specific filters.** We visualize the learned filters of CSR trained with linear classification layers on different datasets. From top to bottom, we present combined filters in Fourier space, individual filters in Fourier space, and individual filters in image space. Filters optimized for CIFAR 10/100 and xView have higher spectral energy in the low-frequency regions, while filters optimized for the MRI Patch dataset focus more on high-frequency regions.

MRI Patch (500 samples) datasets. As shown in Figure 4, the filters optimized for all four datasets present wider bandwidths than the initial filters, resulting in better coverage of the Fourier domain.

When comparing the filters optimized for different datasets, we notice that the filters designed for MRI Patch present an even **higher concentration of high-frequency energy**, whereas the filters for CIFAR 10/100 and xView focus more on l**ow-frequency regions**. This observation implies that the classification of MRI patches heavily relies on high-frequency details, while others are more sensitive to low-frequency features.

## 4.3 ABLATION STUDIES

We evaluate the contributions of learnable filters and our proposed H-CReLU in CSR through ablation studies. We report the results on CIFAR 10 (full size) and MRI patch classification (100 samples) for CSR + LL and CSR + CDS type-I. More results can be found in the supplementary. Our experimental setup for CSR (Table 3) includes the following configurations: 1) fixed filters and

| Method | L. F. | H-C. | CIFAR 10 | MRI Patch |
|---|---|---|---|---|
| CSR + LL | - | - | 66.02 | 58.16 |
| | ✓ | - | 71.23 ↑5.21 | 68.85 ↑10.69 |
| | - | ✓ | 70.35 ↑4.33 | 70.07 ↑11.91 |
| | ✓ | ✓ | **74.30 ↑8.28** | **74.22 ↑16.06** |
| CSR + CDS[†] | - | - | 74.51 | 62.55 |
| | ✓ | - | 77.60 ↑2.89 | 74.03 ↑11.48 |
| | - | ✓ | 79.02 ↑4.51 | 78.40 ↑15.85 |
| | ✓ | ✓ | **81.52 ↑7.01** | **84.80 ↑22.25** |

[†]: CDS type-I (Singhal et al., 2022a); L.F.: Learnable filters; H-C.: High-dimensional C-ReLU (H-CReLU)

Table 3: **Ablation studies of different CSR components.** We analyze the contributions of learnable filtering and H-CReLU for CSNs on CIFAR 10 and MRI patch benchmarks.

| Method | Activation | CIFAR 10 | MRI Patch |
|---|---|---|---|
| CSR +LL | Modulus | 71.23 | 68.85 |
| | CReLU | 70.88 ↓0.35 | 65.33 ↓3.52 |
| | GTReLU | 71.04 ↓0.19 | 69.08 ↑0.24 |
| | **H-CReLU (Ours)** | **74.30 ↑3.07** | **74.22 ↑5.37** |
| CSR +CDS | Modulus | 77.60 | 74.03 |
| | CReLU | 75.08 ↓2.52 | 71.01 ↓3.02 |
| | GTReLU | 78.24 ↑0.64 | 76.40 ↑2.37 |
| | **H-CReLU (Ours)** | **81.52 ↑3.92** | **84.80 ↑10.07** |

Table 4: **Ablation studies of different nonlinear activation functions with learnable filters.** We compare our H-CReLU with other complex-valued activation functions. H-CReLU yields the best results. ↑ and ↓ indicate an increase and decrease in classification accuracy, respectively.

complex modulus as activation function $(-,-)$; 2) learnable filters and complex modulus $(\checkmark,-)$; 3) fixed filters and H-CReLU $(-,\checkmark)$; and 4) learnable filters and H-CReLU $(\checkmark,\checkmark)$.

Table 3 demonstrates that integrating learnable filters and H-CReLU into CSR results in enhanced performance with minimal parameter increase. Tabel 4 further compare H-CReLU with other complex-valued activation functions: 1) Complex modulus; 2) Complex ReLU (CReLU); 3) learn-able Generalized Tangent ReLU proposed in Singhal et al. (2022a). To ensure fairness, we keep the learnable filter module for all the experiments. Our findings suggest that CReLU is not as effec-tive as modulus in producing higher accuracy due to the phase information loss. GTReLU slightly outperforms modulus in certain experiments. In comparison, H-CReLU yields the most significant improvement compared to other methods, demonstrating its superiority as a non-linear activation module for CSR.

### 4.4 CSR FOR FEW-SHOT LEARNING

Few-shot learning is a popular machine learning sub-field that aims to train models capable of rec-ognizing and classifying new objects or categories with only a few examples or instances. In this section, we evaluate the effectiveness of CSR for few-shot learning on the CIFAR 10 dataset and compare its performance with S (Bruna & Mallat, 2013), LS (Gauthier et al., 2022) and CDS. We be-gin by training CSR and other models on images from 5 subclasses in the CIFAR 10 dataset, which include 25,000 training images from the following classes: airplane, automobile, bird, cat, and deer. Next, we fine-tune the models on few-shot images (5 and 10 samples from each class) from the re-maining 5 classes: dog, frog, horse, ship, and truck. Finally, we evaluate the classification accuracy of the fine-tuned models on images of the second set of 5 classes (2,500 images).

| Method | CIFAR 10 | |
|---|---|---|
| - | 5 samples | 10 samples |
| **_Scattering + Linear layers_** | | |
| S (Bruna & Mallat, 2013) + LL | 52.70±3.01 | 64.56±1.27 |
| LS (Gauthier et al., 2022) + LL | 53.52±3.33 | 66.16±1.32 |
| CSR + LL † | **55.62**±2.94 | **68.74** ±1.48 |
| **_Scattering + CIFARNet_** | | |
| S (Bruna & Mallat, 2013) + CIFARNet | 58.49±2.61 | 66.60±2.70 |
| LS (Gauthier et al., 2022) + CIFARNet | 58.95±3.30 | 65.45±1.90 |
| CSR + CDS type-I † | **60.12** ±2.53 | **68.04**±1.38 |
| **_CDS type-I_** | | |
| CDS type-I (Singhal et al., 2022a) | 51.34±3.22 | 59.91±2.02 |

† ours ; S: Scattering; LS: Learnable Scattering; CSR: Complex-valued Scattering Representations (ours).

Table 5: **Few-shot classification results on subset of CIFAR 10.** We pre-train the models on 25,000 images from 5 subclasses in CIFAR 10. Next, we fine-tune the models on few-shot images from the remaining 5 classes and evaluate the testing images (2,500) of the second set of 5 classes.

Table 5 summarizes the results for both the 5 samples and 10 samples experiments. It can be ob-served that CSR outperforms its real-valued counterparts and CDS. Moreover, CSR+CDS outper-forms CDS by 8.78% and 8.13% in the 5 and 10 samples few-shot learning experiments, respec-tively, which highlights the potential of CSR for few-shot learning.

## 5 CONCLUSION

In this work, we propose Complex-valued Scattering Representations (CSR) as a novel and universal complex-valued representation for a wide range of input domains, including RGB, MRI, and MSI, in the field of complex-valued deep learning. The incorporation of tunable data-specific wavelet filters and H-CReLU enables CSR to effectively capture both spatial and spatial-frequency properties of input data. By integrating CSR into complex-valued models for image classification, we have achieved significant performance gains compared to real-valued counterparts and complex-valued models without CSR, especially under _limited labeled_ training data settings. Therefore, CSR can greatly enhance complex-valued networks on a broader range of applications.

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
