# OpenReview forum: "Complex-valued Scattering Representations"
_ICLR.cc/2024/Conference — ICLR 2024 Conference Withdrawn Submission_

### Official Review · Reviewer_sNU1 · 2023-10-16

**Soundness:** 2 fair
**Presentation:** 4 excellent
**Contribution:** 2 fair
**Rating:** 5
**Confidence:** 3

**Summary:**

The paper proposes Complex-valued Scattering Representations (CSR) as universal complex-valued representations and integrate
them into complex-valued deep learning networks. The network contains complex-valued Morlet wavelets with tunable parameters to model the spatial and spatial-frequency properties of the input data, and learnable high-dimensional complex-valued ReLU as the nonlinear activation function. The model significantly outperform real-valued counterparts and existing complex-valued models on a variety of image classification tasks, especially under limited labeled training data settings.

**Strengths:**

- The article is very well written and a pleasure to read.

- The architecture of the proposed model is clearly depicted in the figure and easy to follow.

- Promising results are achieved in a comprehensive experiment, where the proposed model consistently beats the SOTA by under both full-dataset and few-shot settings on different image classification tasks.

**Weaknesses:**

- The contribution of the paper, albeit original, is rather incremental, because it mainly combines an existing work for constructing complex-valued input and another for building learnable filters.

- The work is highly related to to Gauthier et al. 2022, but there is a lack of discussions on why this model is better than  Gauthier et al. 2022 on a conceptual level, and how it is reflected in the experiment result.

- Even though CSR beats the SOTA on few-shot learning, there are no discussions that shed light on why it is so.

**Questions:**

Even though the proposed model achieves promising results, it is unclear to me the theoretical contributions of this paper. Answers to the following questions could help me better understand the value of the paper:

- Are there any high-level reasons underlying the combination of sliding encoding and wavelet transform? Are there any potential fitness of the wavelet transform to the sliding-encoded input?

- What are the crucial elements of this model that makes it superior to Gauthier et al. 2022's work? Can the superiorities be seen from the learned filters?

- Are there any insights on why this model has high few-shot learning capability, apart from the empirical evidence that scattering networks excel under limited labeled training data settings?

---

### Official Review · Reviewer_G9gH · 2023-10-19

**Soundness:** 2 fair
**Presentation:** 2 fair
**Contribution:** 2 fair
**Rating:** 5
**Confidence:** 4

**Summary:**

This paper proposes to feed complex-valued deep learning networks with coefficients extracted of Complex-valued Scattering Representations (CSR) of input data when dealing with either complex or real input samples in a supervised learning scenario. CSR consists on the application of a sequence of layers composed by convolutions with Morlet filters with learnable parameters followed by a novel learnable activation function. The complex-valued coefficients obtained at each CSR layer are subsequently fed into traditional complex-valued deep learning models. Empirical findings demonstrate that the introduction of CSR can lead to improved classification performance, particularly when dealing with a limited number of training data samples.

**Strengths:**

- The main strength of this study lies in its provision of an effective decomposition of a real- or complex-valued signal. This decomposition enables successful classification by deep complex-valued neural networks, which have gained prominence in recent literature due to advances in manifold geometry and group theory.

- Another notable contribution is the introduction of a novel complex-valued activation function equipped with learnable parameters, which has demonstrated empirical improvements in classification performance.

- The experimental results presented in this work convincingly showcase the superiority of the proposed approach when compared to conventional complex-valued deep neural networks without using the proposed CSR.

**Weaknesses:**

Main issues

-	Although there is an intuitive understanding of why the suggested CSR offers improved data representation for classification, the paper falls short in terms of providing a solid theoretical foundation to comprehensively explain its underlying principles and limitations.

-	The contributions outlined in the paper signify an evolutionary enhancement of previously introduced complex-valued deep neural networks.

Minor issues

-	Before eq. (3): “w is the learnable parameters” ->  “w is the vector of learnable parameters”

**Questions:**

I notice that a particular linear transformation of real-valued input data is applied to convert it into a complex-valued input (sliding color encoding proposed in Singhal et al., 2022a). I have a few questions regarding this selection:

1.	Given the numerous possible methods to transform real-valued data into complex-valued inputs, why was this particular transformation chosen as the preferred option?

2.	Is there a conceptual or empirical rationale behind the adoption of this specific transformation?

3.	Have any attempts been made to introduce a trainable linear transformation, allowing it to be adjusted during training? Do you believe that this approach might offer any advantages?

---

### Official Review · Reviewer_X1jj · 2023-10-30

**Soundness:** 4 excellent
**Presentation:** 4 excellent
**Contribution:** 3 good
**Rating:** 6
**Confidence:** 4

**Summary:**

Complex-valued scattering representations are proposed as a more effective representational space construction.  Complex-valued machine learning models (e.g. complex-valued classifiers) are then poposed as a second stage to address the task at hand.  While the scattering network is largely designed, some of its hyperparameters are learned during training.  Furthermore, a learnable pointwise activation function is introduced, which first lifts to a higher dimensional space through a learned projection, performs a complex ReLU, before a learned down projection to the original space.  The motivation of the general approach is not simply for handling complex-valued data but rather to provide a more effective representational space, even for real-valued data.  A number of experiments are performed, both for real- and complex-valued data, demonstrating that the introduced method achieves superior performance to previous approaches.  Ablation studies to demonstrate the performance of various components are also performed.

**Strengths:**

The introduced complex-valued scattering representation provides a more effective representation space for subsequent learning tasks (e.g. classification), which can then be coupled with other complex-valued deep learning models.  Beyond the the usual scattering approach, an end-to-end training approach is considered, which allows some of the hyperparameters of the scattering network to be simultaneously learned (e.g. wavelet orientations and aspect ratios), along with the learned activation funcrtions.  Extensive benchmark experiments are performed, demonstrating superior performance to existing techniques.  Relatively extensive ablation studies are also presented.

**Weaknesses:**

The introduced methodological contributions are relatively straightforward.  A real-valued wavelet has simply been replace by a complex-valued wavelet and the non-linear activation function of the scattering network switched from a modulus to complex ReLU.  In addition, I admit the authors do introduce a learned complex ReLU.  It would also be helpful if complex ReLUs could be briefly described, either in the main body or appendix, so that the paper is self-contained.

I am not convinced that the proposed complex scattering network is stable to deformations and invariant to local transformation as is claimed in the final paragraph of page 4, where it is claimed that proofs presented in prior works also hold here.  I belive the proofs of these results rely on the fact the the modulus operator adopted for the activation functions in alternative scattering networks is non-expansive.  Since the activation function is replaced with a learned complex ReLU, it is not clear to me that the resulting operator would be non-expansive.  If the authors are to claim these theoretical properties also hold for their approach then this needs to be considered in more depth.

While learning some of the parameters of the scattering representation and the activation functions in an end-to-end training manner (as discussed in Strengths above) indeed has the advantages that a more effective representation space can be found, it has the disadvance that the computation of the scattering representation cannot be decoupled from training, resulting in training that is more computationally expensive.  This should be commented.

**Questions:**

- As discussed in Weaknesses, please consider the stability and invariance properties on the scattering network in greated detail.  It is not clear to me that these results necessarily hold here.  Perhaps they do (indeed, the proposed scattering networks acheive excellent perfornace for downstream tasks, e.g. classificaiton) but if so that needs to be elaborated in greater detail.  Alternatively, this claim could also be dropped if it doesn't hold.

- Is a complex encoding of real-valued data actuall necessary (as specified in Equaiton 1)?  Since a complex-valued wavelet is consided, wavelet coefficients will still be complex-valued, even if the signal/data is real.

- Scattering networks were first presented in [Mallat (2012)](https://onlinelibrary.wiley.com/doi/10.1002/cpa.21413), which should be made clear and the paper cited.  I believe Bruna & Mallat (2013) presented the first application of scattering networks to classification but did not introduce scattering networks per se.

- The references should be reviewed to ensure reference details are up-to-date.

---

### Official Review · Reviewer_yQE4 · 2023-10-30

**Soundness:** 3 good
**Presentation:** 1 poor
**Contribution:** 2 fair
**Rating:** 3
**Confidence:** 3

**Summary:**

The paper proposes a complex valued scattering representation and a new complex activation function the H-CReLU. The proposed architecture is evaluated in the CIFAR 10, CIFAR 100 and an MRI patch classification dataset. The experimental section reports improved performance on smaller subsets of the training data.

**Strengths:**

- To my knowledge, the proposed activation function is novel. It has not been introduced in this form.
- The paper's general direction is exciting and interesting.
- The experimental section reports improved performance on smaller subsets of the training data.

**Weaknesses:**

- Some of the salesmanship in the introduction is very hard to understand.

- In addition to [1], complex ReLUs have previously appeared in [2] and [3]. Both [2] and [3] are not cited. It felt is was misleading that [1] is cited as " (Trabelsi et al., 2017) redefine basic building blocks for complex-valued networks, such as complex-valued convolution, batch normalization, and non-linear activations." While the description is technically correct, [1] introduces the cReLU. Since this paper also claims the introduction of a new ReLU as its contribution, citing [1]'s contribution as having proposed some non-linear activation in the related work section is too vague.

- Previous work by [4] on the complex dual-tree wavelet transform in deep learning introduces the DTCWT (Dual-Tree Complex Wavelet Transforms)-Scatter net. This seems to be precisely what this paper is doing. Unfortunately, [4] is not discussed in the current version of this draft.

- The experimental section is limited to comparatively small data sets.


[1] C. Trabelsi, O. Bilaniuk, Y. Zhang, D. Serdyuk, S. Subra-
manian, J. F. Santos, S. Mehri, N. Rostamzadeh, Y. Bengio,
and C.J. Pal. “Deep Complex Networks.” In: ICLR. 2018.

[2] Arjovsky, Martin, Amar Shah, and Yoshua Bengio. "Unitary evolution recurrent neural networks." International conference on machine learning. PMLR, 2016.

[3] N. Guberman. On Complex Valued Convolutional Neural
Networks. Tech. rep. The Hebrew University of Jerusalem
Israel, 2016.

[4] Uses of complex wavelets in deep convolutional neural networks
F Cotter - 2020 - repository.cam.ac.uk

**Questions:**

- The introduction claims current complex representations are limited to modelling the channel dimension. Why would a possibly three-dimensional complex convolutional network model, i.e. as introduced by Trabelsi et al., only model only the channel dimension? The wavelet transform is based on the convolution operation. If a fixed wavelet filter can model multichannel interactions, why can't the learnable kernels from Trabelsi et al.?

- Have other existing complex activation functions, like the Hirose-activation [5], been considered?

- The first sentence of the abstract promises manifold geometry and group theory. Is there some in the paper?

[5] Complex-valued neural networks, A Hirose, Advances and Applications, 2013 Wiley Online Library